# A Colorimetric Nanofiber Film Based on Ethyl Cellulose/Gelatin/Purple Sweet Potato Anthocyanins for Monitoring Pork Freshness

**DOI:** 10.3390/foods13050717

**Published:** 2024-02-27

**Authors:** Peng Wen, Jinling Wu, Jiahui Wu, Hong Wang, Hong Wu

**Affiliations:** 1College of Food Science, Guangdong Provincial Key Laboratory of Food Quality and Safety, South China Agricultural University, Guangzhou 510642, China; wujinling331@163.com (J.W.); gzwhongd@163.com (H.W.); 2School of Food Science and Engineering, South China University of Technology/Guangdong Province Key Laboratory for Green Processing of Natural Products and Product Safety, Guangzhou 510641, China; wujiahui980814@163.com

**Keywords:** electrospinning, colorimetric nanofiber film, pH and ammonia sensitive, freshness monitor

## Abstract

In this study, colorimetric indicator nanofiber films based on ethyl cellulose (EC)/gelatin (G) incorporating purple sweet potato anthocyanins (PSPAs) were designed via electrospinning technology for monitoring and maintaining the freshness of pork. The film presented good structural integrity and stability in a humid environment with water vapor permeability (WVP) of 6.07 ± 0.14 × 10^−11^ g·m^−1^s^−1^Pa^−1^ and water contact angle (WCA) of 81.62 ± 1.43°. When PSPAs were added into the nanofiber films, the antioxidant capacity was significantly improved (*p* < 0.05) with a DPPH radical scavenging rate of 68.61 ± 1.80%. The nanofiber films showed distinguishable color changes as pH changes and was highly sensitive to volatile ammonia than that of casting films. In the application test, the film color changed from light pink (fresh stage) to light brown (secondary freshness stage) and then to brownish green (spoilage stage), indicating that the nanofiber films can be used to detect the real-time freshness of pork during storage. Meanwhile, it could prolong the shelf life of pork by inhibiting the oxidation degree. Hence, these results suggested that the EC/G/PSPA film has promising future for monitoring freshness and extending shelf life of pork.

## 1. Introduction

With the growing concern of food quality and safety among consumers, colorimetric film, being capable of monitoring the real-time freshness of fresh meat, has aroused increasing attention of researchers. Owing to the generation of the characteristic metabolites of total volatile basic nitrogen including ammonia (NH_3_), dimethylamine (C_2_H_7_N), and trimethylamine (C_3_H_9_N) during the spoilage of fresh meat [1], a basic environment in the packaging environment was formed. As a result, pH-sensing colorimetric films can be exploited to detect the pork freshness by sending colorimetric signals. Recently, an anthocyanin-based sensing indicator has shown great potential due to its non-toxicity [2], widespread sources [3], and good antioxidant property [4]. Purple sweet potato anthocyanins (PSPAs) are such a kind of pigment, showing obvious color changes from acidic to basic environment [5]. Moreover, a study has reported that the higher proportion of acylated anthocyanins in PSPAs made it suitable to be used as indicator agent in intelligent film due to its enhanced stability [6]. For example, a recent study described that a sodium alginate-based pH-sensitive film loaded with PSPAs was designed via casting technique, and its color changed from light pink to light blue, possessing a limited color response [7]. In another study, an indicator film composed of carboxymethyl-cellulose/starch and PSPAs was developed, the color of which varied from light-red to grey-blue, which was difficult to be observed by the naked eye [8].

With the development of nanotechnology, nanomaterials exhibit great advantages for the fabrication of colorimetric film due to the larger surface area that would be more sensitive to the color response [9,10]. In this regard, electrospinning stands out for its simplicity and flexibility to develop a nano-structured film involving neither high-temperature nor high-pressure conditions [11], which is beneficial for maintaining the bioactivity of anthocyanins [12]. Moreover, the advantages of electrospun nanofibers on enhancing the sensing abilities for pH changes compared to conventional films were also emphasized in recently studies [13,14], highlighting its great potential for the preparation of intelligent materials. For instance, a visual film composed of polyvinylidene fluoride fibrous material and blended with roselle anthocyanins was fabricated. The value of R, G, and B changed rapidly within 30 min, indicating that the sensitivity of colorimetric film was efficient and responsive to the volatile ammonia [9].

Owing to the concern for environmental protection, biodegradable, and sustainable biopolymers are preferred over synthetic polymers [15,16]. In particular, considering the limited ability of hydrophilic films, ethyl cellulose (EC) stands out as a suitable choice for the preparation of colorimetric film owing to its nontoxicity, hydrophobicity and good property of fiber-forming, which is beneficial for enhancing the stability in high humidity environment [17]. On the other hand, researchers have found that the hydrophobic property was unfavorable for the sensitivity of indicator [9]. Thus, a challenge arises in balancing the stability and sensitivity of the colorimetric film. To address this, gelatin (G), was induced into the matrix to enhance the hydrophilicity of EC-based film due to its high hydrophilic nature. A previous study by Liu and colleagues indicated that a hydrophilic surface was observed in the electrospun EC/G film, which had a higher gelatin ratio of 75%, with a water contact angle of 53.5° [18]. Although studies for the colorimetric films loaded with PSPAs had been reported [5,8,19], they were fabricated by the casting method. To our knowledge, there is no study on the preparation of PSPA-loaded colorimetric nanofiber film based on biodegradable materials of EC and G by electrospinning.

Hence, this study aimed to encapsulate PSPAs into the EC/G matrix via electrospinning for developing novel colorimetric film. The morphology and structure characterization and other physical properties were observed by scanning electron microscope (SEM), Fourier transform infrared (FTIR), water contact angle (WCA), and water vapor permeability (WVP), respectively. The multifunctional colorimetric nanofiber films were evaluated by investigating pH sensitivity, volatile ammonia response and antioxidant properties. Moreover, the colorimetric film was subjected to the pork freshness monitoring.

## 2. Materials and Methods

### 2.1. Materials

Purple sweet potato powder was bought from the Zhuangmu Food Co., Ltd. (Shandong, China). EC (3–7 mPa·s, 5% toluene/isopropyl alcohol 80:20), hydrochloric acid (HCl), acetic acid, disodium hydrogen phosphate, citric acid, ammonia, and 2,2-diphenyl-1-picrylhydrazyl (DPPH) were obtained from Macklin (Shanghai, China). G and ethanol were acquired from Aladdin (Shanghai, China).

### 2.2. Preparation of PSPAs

The PSPAs were prepared from purple sweet potato powder. Firstly, 50 g purple sweet potato powder underwent dissolution in a hydrochloric acid-ethanol solution system of 500 mL, with a volume ratio of 15:85. Then, the solution was placed in the ultrasonic treatment (Ningbo Xinzhi Biotechnology Co., Ltd., Ningbo, China) at 50 °C, approximately 270 W for a duration of 30 min. Subsequently, a centrifugation process was carried out at 3000 rpm for 15 min, resulting in the purification of the solution. To eliminate remaining solvents, the purified solution was concentrated in a vacuum rotary evaporator at 45 °C. After freeze-drying for 2 days, PSPA powder was prepared. The PSPA powder should be stored in the dark at 4 °C.

### 2.3. Characterization of PSPAs

A certain quantity of PSPA solution (2 g/L) was added to the buffer solutions, which was composed of disodium hydrogen phosphate (0.2 M), citric acid (0.1 M), HCl (0.2 M), and NaOH (0.2 M). The spectra of the mixture were characterized by a UV-Vis spectrophotometer (Shimadzu, Tokyo, Japan) within the range of 400–700 nm.

The antioxidant capacity of the PSPA solution were evaluated according to Fernandez-Marin et al. with some modifications [20]. Here, 1 mL of PSPA solution was added into 2 mL 0.1 mM DPPH- ethanol solution, after which the mixture was incubated in complete darkness for a duration of 30 min. The samples of the solution from the mixture were taken and the absorbance values were read at 517 nm by the spectrophotometer. The DPPH free radical scavenging activity (%) was determined using Equation (1):Scavenging activity (%) = (A_0_ − A_30_)/A_0_ × 100(1)
where A_0_ is the absorbance of the control and A_30_ is the absorbance of the sample after 30 min.

### 2.4. Preparation of the Nanofiber Films

To fabricate the purple sweet potato anthocyanin-loaded nanofiber films, EC and G with a total mass fraction of 25 wt% and a mass ratio of 1:4 (*w*/*w*) were dissolved in the mixed solvent which contained acetic acid, ethanol, and water at a 3:1:1 (*v*/*v*/*v*) ratio. Afterwards, different content of PSPAs (20, 25, 30, 35, and 40 wt%) based on the total mass fraction of the substrates were added into the mixture and named as ECG−P5.00, ECG−P6.25, ECG−P7.50, ECG−P8.75, ECG−P10.00, respectively. After stirring, film solutions were prepared. The nanofiber films were prepared by using electrospinning technology, which was conducted at a temperature of 25 °C and a relative humidity of 50%. The electrospinning processing parameters were set as follows: applied voltage of 17 kV, flow rate of 0.5 mL/h, and tip-collector distance of 14 cm, using the stainless steel needle with gauge 21 with a diameter of 0.82 mm. Meanwhile, the corresponding ECG-P7.50 casting film was prepared by using the same ECG−P7.50 spinning solution through a casting method. In this case, the film-forming solution (12.5 mL) was poured into a Petri dish with a diameter of 9 cm. The dish was left to dry at room temperature (23 ± 2 °C) for 2 days.

### 2.5. Color Response to Violate Ammonia

The volatile ammonia response of the nanofiber films were measured according to Lin et al. [21] with slight adjustments. Nanofiber films were attached in 50 mL containers containing 50 μL of ammonia solution at room temperature. The ammonia response of the nanofiber films was recorded using a camera for 5 and 10 min. To guarantee the validity of the results, the color changes in nanofiber films were captured in a LED portable photography box with stable light. Afterwards, a digital colorimeter (WSC-2B, Shanghai Wuguang Instrument Co., Ltd., Shanghai, China) was used to extract the color parameters (L*, a*, b*) from the image. And a color extraction (Version 4.0, Shanghai, China) software was used to convert L*, a*, b* value into relative color image.

To assess the film’s sensitivity to ammonia, ΔE (total color difference) was obtained according to the following Equation (2):(2)$$\Delta E=\sqrt{{\left(L^{*}-L_{0}\right)}^{2}+{\left(a^{*}-a_{0}\right)}^{2}+{\left(b^{*}-b_{0}\right)}^{2}}$$
where L_0_, a_0_, b_0_ are the initial color values of the film, L*, a*, b* are the color values after immersion.

### 2.6. Characterizations of Nanofiber Films

#### 2.6.1. Encapsulation Efficiency (EE) of PSPAs

Firstly, calibration curve of PSPAs (R^2^ = 0.9992) was prepared by recording the absorbance of different PSPA solutions at 530 nm. After that, 10 mg nanofilm was immersed into 3 mL of deionized water, and the free PSPA content on the surface of nanofilm was measured by detecting the solution’s absorbance at 530 nm, which can be regard as m_0_. Then, 7 mL of acetic acid aqueous solution (60%, *v*/*v*) was added to completely dissolve the nanofilm, by which, both of free and encapsulated PSPAs were in the solution and the m can be detected spectrophotometrically at 530 nm. Finally, the EE of PSPAs into nanofilm was determined by Equation (3):EE (%) = (m − m_0_)/m × 100(3)
where m is the total content of PSPA encapsulated into the film (g), and m_0_ is the PSPA content on the surface of film (g).

#### 2.6.2. Scanning Electron Microscopy (SEM)

The nanofiber films measuring 6 mm in diameter were placed onto the platform. For purpose of increasing the conductivity, each film was fixed on the carrier platform with conductive adhesive, and the samples were then sputter-coated with gold for 60 s, to achieve a gold film thickness of approximately 30 nm using an automatic coater (Agar Auto Sputter Coater B7341, Agar Scientific Limiter, Stansted, UK). The morphology of nanofiber films was detected on a scanning electron microscopy (SU 5000, Hitachi, Japan) with a magnification of 8.0 k. And the images were made under an accelerating voltage of 3.0 kV. Briefly, 5 SEM images were taken, followed by the random selection of 60 fibers from each image, and the diameter was quantified using Image J (Version 1.40, Bethesda, MA, USA) software package.

#### 2.6.3. Attenuated Total Reflection-Fourier Transform Infrared (ATR-FTIR)

The component interaction in the nanofiber films were assessed by using a FTIR spectrometer (TENSOR 37, Bruker Co., Ettlingen, Germany) in a wavenumber range of 4000–400 cm^−1^.

#### 2.6.4. Water Vapor Permeability (WVP) and Water Contact Angle (WCA)

The determination of WVP was conducted based on the prior literature, incorporating a minor modification [9]. To measure the thickness of nanofiber films, ten random locations were assessed using a vernier caliper (0–100 mm, Hengliang Measuring Tools Co., Ltd., Shanghai, China). Afterwards, the centrifuge tube with dry silica gel was sealed by a test film, after which the tube was stored at 25 °C in oven containing distilled water. The weight of the film was regularly monitored every 24 h for a duration of 7 days. The calculation of WVP was accomplished utilizing the formula provided as follows (4):WVP = (Δm × x)/A × ΔP × t(4)
where Δm is the increased weight of silica gel (g), x is the thickness of the film (m), A represents the permeability area (m^2^) for the water vapor, ΔP is the water vapor pressure between the tube and environment (3179 Pa), and t is the time intervals (s).

The WCA was recorded by a water contact angle analyzer (OCA 40 Micro, Data Physics Co., Filderstadt, Germany). To measure the WCA, 4 μL distilled water was applied to the surface of the film being tested. Subsequently, the film was captured in photographs for a duration of 10 s.

#### 2.6.5. Antioxidant Capacity of Nanofiber Films

Nanofiber films (10 mg) were dissolved in 7 mL of ethanol for 24 h at room temperature. And antioxidant capacity of nanofiber films was measured as descripted in Section 2.3.

### 2.7. Application Test

The nanofiber films were affixed onto the surface of the inner top of the Petri dish. Pork samples derived from tenderloin were placed at 4 °C for 8 days, and the changes were recorded every 2 days. The color parameters (L*, a*, b* and ∆E) were determined as described in Section 2.5. Thiobarbituric acid reactive substances (TBARS) of pork was determined according to the modified method described in the previous research by Ma et al. [22]. Pork (5 g) was mixed with 15 mL of trichloroacetic acid (7.5 wt%). After filtration, 0.02 mol/L thiobarbituric acid solution was added into supernatant. After cooling in 90 °C water bath for 30 min, the absorption at 532 nm was recorded. The TBARS value of pork sample was calculated according to Equation (5):TBARS = (A_532 nm_/m) × 9.24(5)
where A_532 nm_ is the absorbance of the supernatant and m is the weight of pork sample, 9.24 is a constant that was derived from the sample dilution factor and the molar extinction coefficient (1.56 × 10^5^ M^−1^ cm^−1^) of the red TBA reaction product [22].

### 2.8. Data Analysis

The experiments were carried out in triplicate and reported as the mean ± standard deviation. One-way analysis of variant (ANOVA) was employed to analyze the figures using SPSS software (Version 27.0, Chicago, IL, USA). All the graphs were generated using the Origin (Version 9.0, Northampton, MA, USA) software package. For comparing the significant differences, Duncan method at a significance level at *p* < 0.05 was employed.

## 3. Results and Discussion

### 3.1. Color Changes and UV-Vis Spectroscopy of PSPAs

The color and absorption spectra changes in PSPAs under different pH buffer solutions are shown in Figure 1, which was ascribed to the chemical structure transformation of PSPAs. To be specific, at pH 1–3, the PSPA solution was in a pink color as the stable flavylium cation form predominated [23]. A colorless carbinol pseudo base formed, which was the reason for the color turned pale pink when the pH rose from 4 to 6. While at pH 7, quinonoidal base was induced by a rapid deprotonation, echoing the color turned into purple [24]. As the pH value increased (8–11), the PSPAs were converted into a blue anionic quinonoidal base, and even mild green or yellow colors appeared for pH 12 and 13, respectively, which were attributed to the formation of chalcone by opening the pyran ring of anthocyanin.

Of note, the absorption peak of PSPA solutions also exhibited significant shifts as the pH continuously changed. When the pH was below 3, the maximum absorbance at 530 nm was observed. In the alkaline condition, the absorption peak was shifted to 610 nm, which was consistent with the color changes from pink to blue. Wang et al. also reported a similar result, as the absorption peak of eggplant anthocyanins shifted from 540 nm to 580 nm in different pH buffer solutions [25]. Therefore, attributed to structural transformation in PSPAs, the solution displayed a visible color change, demonstrating its potential as a natural pH-sensing dye.

### 3.2. Colors and Antioxidant Properties of PSPAs

According to the above results of PSPA color changes, by incorporating PSPAs into ECG, the obtained colorimetric nanofiber films were then exposed to volatile ammonia. From Figure 2a, it was noticeable that the pink to green color changes for nanofiber films were easily captured, and the higher content of PSPA, the more obvious color responses. Similarly, as for DPPH radical scavenging rates, good antioxidant property of PSPAs was also observed owning to the abundant phenolic hydroxyl groups. Considering the price-performance of the PSPA ratio and the film’s final color that was exposed to volatile ammonia, ECG−P6.25, ECG−P7.50, and ECG−P8.75 nanofiber films were further studied in the following experiments.

### 3.3. EE and Morphology Analysis

The EE values of PSPA in ECG−P6.25, ECG−P7.50, and ECG−P8.75 were 78.80 ± 2.06%, 83.05 ± 1.59%, and 83.20 ± 0.85%, respectively, indicating that PSPAs can be effectively encapsulated via electrospinning technique. As shown in Figure 3b–d, all the surface of nanofiber films were smooth and homogeneous without any beads, while the average diameter experienced an augmentation from 519 nm to 590 nm with the increasing PSPA content. The phenomenon was attributed to the increased viscosity of the solution when PSPAs were added, making it harder for the electrospinning jet to stretch, thus resulting in a larger fiber diameter [9]. Similar morphology was also reported in a previous study [26].

### 3.4. FTIR Spectroscopy of the Film

The location and strength of the infrared absorption peak mirror the chemical clusters of various elements and their interplay [26]. As presented in Figure 4a, the O-H stretching vibration peak at 3478 cm^−1^ and the C-H asymmetric stretching vibration peaks at 2977–2875 cm^−1^ were the major peaks of EC. The characteristic peaks of G at 1647 cm^−1^ (amide I), 1539 cm^−1^ (amide II), and 1243 cm^−1^ (amide III) were assigned to C=O, N-H, and C-N stretching vibration, respectively [27]. As for PSPA, the strong absorption at 3397 cm^−1^ was attributed to the O-H stretching vibration of phenolic hydroxyl groups [28], and the peaks at 2939 cm^−1^, 1076 cm^−1^ resulted from the stretching vibrations of C-H and C-O-C [8]. Regarding the FT-IR spectra of EC/G/PSPA, the O-H stretching shifted from 3397 cm^−1^ to 3375 cm^−1^, indicating the creation of the hydrogen bonds between PSPA and polymers. A similar phenomenon was obtained by Guo and his colleagues, noting that the formation of hydrogen bonds between pectin and beetroot extract caused a smaller shift in the peak [29]. Furthermore, the characteristic peak of PSPA was covered in the EC/G/PSPA film, signifying the effective integration of anthocyanins into the films.

### 3.5. Physical Properties of the Film

The film’s wettability and barrier property were crucial for the film’s stability and integrity in humid environments. The static water contact angle (WCA) at a 90° angle is often considered the critical point that determines whether a film surface is hydrophilic or hydrophobic, which can be affected by various factors including the intermolecular hydrogen bonds, the ionization state of amino and carboxyl groups, etc. [30]. From Figure 4b, the WCA of the film increased from 70.88 ± 2.94° to 81.62 ± 1.43° with the addition of PSPA, indicating the formation of a more hydrophobic micro-nano surface. This phenomenon could be linked to the creation of hydrogen bonds between PSPA and certain hydrophilic groups in gelatin (-NH_2_ and -OH), diminishing the interplay of hydrophilic groups on the surface of gelatin and water. Similarly, the increased WCA value of chitosan/chitin nanofiber films was also observed by adding eggplant peel anthocyanins in Wang’s study, from hydrophilic nature to be hydrophobic rough surface [25]. Still, the WCA of our study indicated the low hydrophilicity, which might facilitate the diffusion of H^+^ or OH^–^ into the film, thereby enhancing the color response sensitivity as a pH-sensitive film [31]. WVP is an index of evaluating the film’s ability to reduce moisture transfer between food and environment [32]. With the increase in the PSPA concentration, the WVP of the films decreased significantly (*p* < 0.05). It can be explained that the PSPA strengthened the matrix’s internal network, improving the film compactness and preventing the diffusion of water molecules through the film [33]. As expected, the water affinity of the film could be attenuated due to the formation of hydrogen bonds [34], which was consistent with the WCA results. In line with Wu’s study, the addition of red cabbage extract also reduced WVP of pullulan/chitosan/CN film [35]. However, there was no further decrease for the higher content of 8.75% PSPA, since the aggregation of anthocyanins could destroy the interaction between EC and G, leading to a decrease in compactness and uniformity of the film [36]. Guo et al. found that the WVP of purple cabbage anthocyanins film decreased firstly and then increased with further addition of anthocyanins [37]. Different trends of WVP values may be related the different amounts of anthocyanins. Therefore, the obtained hydrophilic EC/G/PSPA nanofiber film was expected to improve the sensitivity of pH response and possess good moisture resistance, exhibiting a great potential in the high humidity environment.

### 3.6. Antioxidant Capacity of the Film

Oxidation is one of the major causes for the detrimental effects such as color fading, flavor and texture deterioration of meat products, and even reduce consumer acceptance of meat products [38]. Herein, the DPPH radical scavenging activity of the film was evaluated (Figure 4d). The results indicated that the EC/G/PSPA nanofiber film exhibited promising antioxidant activity due to the presence of numerous phenolic hydroxyl groups of PSPA that could capture free radicals and inhibit the free radical chain reactions [39], the obtained EC/G/PSPA nanofiber film possessed good antioxidant activity, with the DPPH radical scavenging activity were 52.53 ± 2.68%, 65.59 ± 0.44% and 68.61 ± 1.80% with the increase in PSPA content, respectively. Similarly, the strong antioxidant activity of anthocyanin extract has been reported [40]. It can be seen that good antioxidant capacity was observed for nanofiber films, ascribing to the increased molecular chemical stability and reduced degradation by the effective encapsulation of phenolic compounds.

### 3.7. Color Response to Violate Ammonia of the Film

Since volatile nitrogen compound can be produced during the spoilage process of meat products, herein, the response to volatile ammonia in vitro was measured via exposing in ammonia solutions for different time. As shown in Table 1, the color of nanofiber films transformed from the initial pink to green and eventually yellow-green, corresponding to the decrease in the a* value and increase in the b* value in the ammonia atmosphere. This change in color was attributed to the absorption of volatized ammonia and water vapor by the fibrous film, resulting in the formation of hydrated ammonia [41], as well as the structure and color changes in the anthocyanins in the ECG−P7.50 nanofiber film. Similar results were observed in other colorimetric films containing eggplant anthocyanins [25], cristata flower extract [42], black rice bran anthocyanins [43], and barberry anthocyanins [44], etc. In addition, it was reported that the color differences among the films can be detected by the human eye when the ΔE values are higher than 5.0, and values exceeding 12.0 indicates a significant distinction between the two colors [45]. It was calculated that the ∆E values of the nanofiber films were 15.58 ± 1.54, 25.45 ± 1.84, and 28.49 ± 1.00, respectively, suggesting that the changed color was noticeable by the naked eye. However, the changes in dark color were difficult to be observed in the casting film, which was not conducive for consumer to identify the freshness of pork by the naked eye. Furthermore, it was observed that the nanofiber films exhibited significantly higher the ∆E value compared to the casting films, indicating that the nanofiber films were more sensitive for freshness monitoring. The response behavior was attributed to the hydrophilicity and the porous structure of nanofiber films, which favored the color reaction by facilitating water molecules and contact sites. Similarly, Maftoonazad et al. discovered that the electrospun nanofiber films based on polyvinyl alcohol and red cabbage anthocyanins can improve the color change response of anthocyanins due to its larger surface area [13]. Therefore, the sensitivity of ECG−P7.50 nanofiber film towards detecting ammonia highlights its potential for monitoring pork freshness.

### 3.8. Application of ECG−P7.50 Nanofiber Film for Pork Freshness

The practical application of obtained colorimetric film was explored to investigate its effect on monitoring pork freshness. As described previously, the presence of PSPAs made it easy to delay the oxidation of pork and enable the surveillance of pork freshness. Thus, the TBARS value was also determined to reflect the oxidation degree of unsaturated fatty acids in meat, as shown in Table 2, the higher the TBARS value, the more severe the oxidation. The initial TBARS value of pork was 0.22 mg/kg. Over time, the TBARS value of pork gradually increased, reaching 0.57 mg/kg on the day 8th day. Compared with the blank group pork (0.66 mg/kg), the ECG−P7.50 group pork showed lower levels of lipid oxidation. In particular, after a storage period of 6 days, the color of the film underwent a transition from pink to light brown, and the film’s color turned brownish green on the 8th day. As expected, the L* value of the film color increased at first for 4 days, then decreased rapidly to 75.63 (*p* < 0.05), the a* value decreased significantly from 7.63 to −0.31 (*p* < 0.05), and the b* value increased markedly from 6.65 to 12.94 (*p* < 0.05), which was consistent with the above color change. The results were in line with the study of Zhang et al., who found that the color changed from grayish-purple to blue-green when exposed to ammonia vapor for anthocyanins-loaded ovalbumin-propylene glycol alginate colorimetric film [46]. Moreover, the ΔE substantially increased (*p* < 0.05), suggesting that the film color changed noticeably during the storage period. In conclusion, the colorimetric film not only extended the shelf life of meat, but also facilitated real-time freshness monitoring.

## 4. Conclusions

Multifunctional EC/G/PSPA colorimetric nanofiber films were successfully prepared via electrospinning. The effective encapsulation and high reaction surface of the film were achieved. FTIR results verified the hydrogen bond interaction between the PSPA and the polymers, and at the same time, the PSPAs endowed the films with favorable wettability and antioxidant activity. Notably, the film exhibited better violate ammonia response sensitivity in comparison with that of casting film. This application study demonstrated that the pH sensitive colorimetric nanofiber films corresponded well with the deterioration of pork, showing a distinct color change from light pink (fresh stage) to light brown (secondary freshness stage) and then to brownish green (spoilage stage), and effectively inhibited the oxidation of pork. Therefore, this study confirmed that the colorimetric film has great potential in monitoring pork freshness in real time and prolonging the shelf life.

## Figures and Tables

**Figure 1 foods-13-00717-f001:**
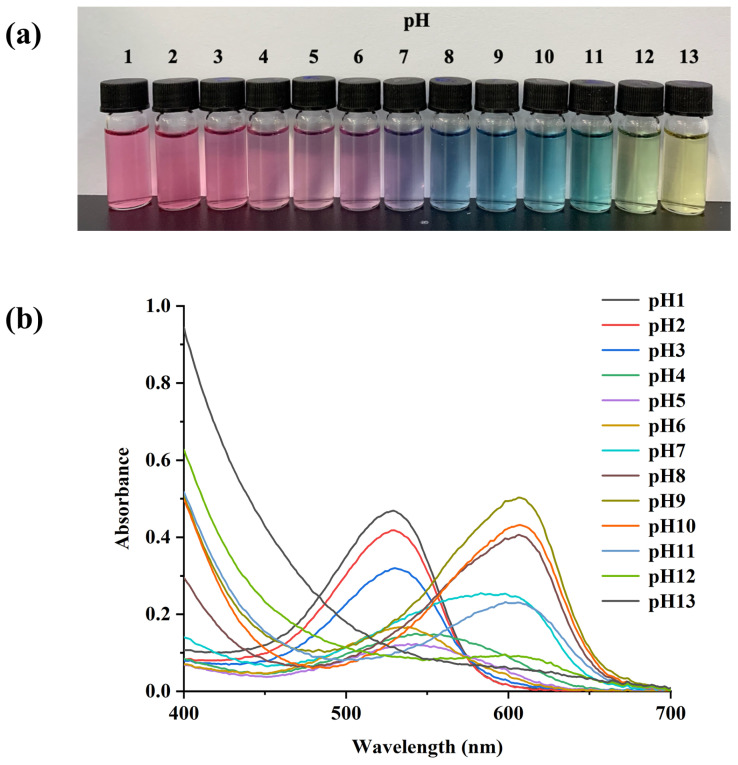
Color changes (**a**) and UV-Vis spectra (**b**) of PSPAs at different pH levels (1–13).

**Figure 2 foods-13-00717-f002:**
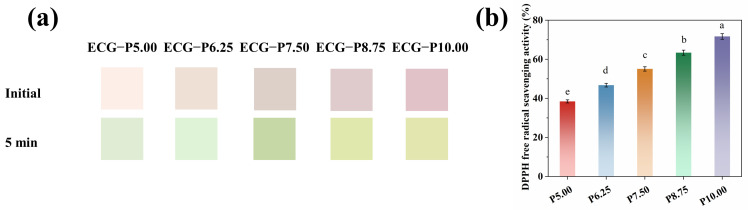
The color of different nanofiber films (initial color and after storage for 5 min) (**a**); DPPH free radical scavenging activity of different concentrations of PSPA solution corresponding to nanofiber films (**b**), respectively. For graphs, the different letters (a–e) indicates that the samples are of significant difference (*p* < 0.05).

**Figure 3 foods-13-00717-f003:**
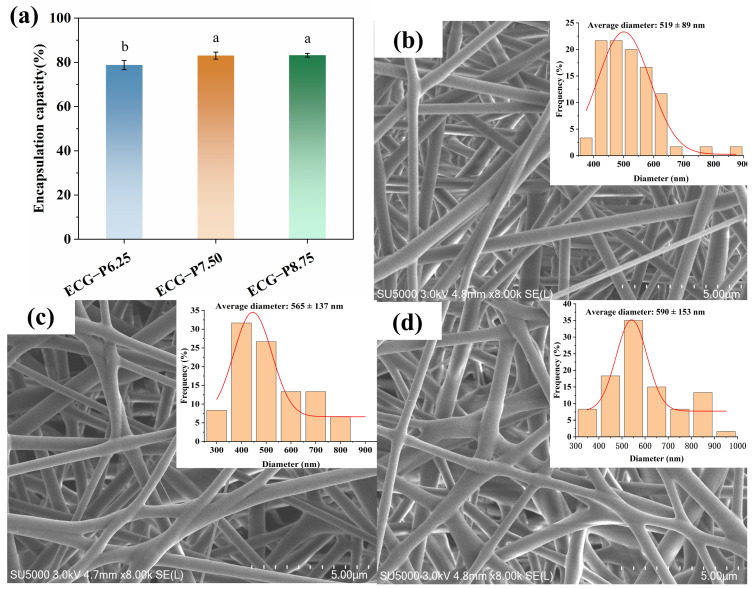
EE of PSPAs into different films (**a**); SEM spectra of ECG−P6.25 (**b**), ECG−P7.50 (**c**) and ECG−P8.75 (**d**), respectively. The same lowercase letters represent no significant difference (*p* > 0.05).

**Figure 4 foods-13-00717-f004:**
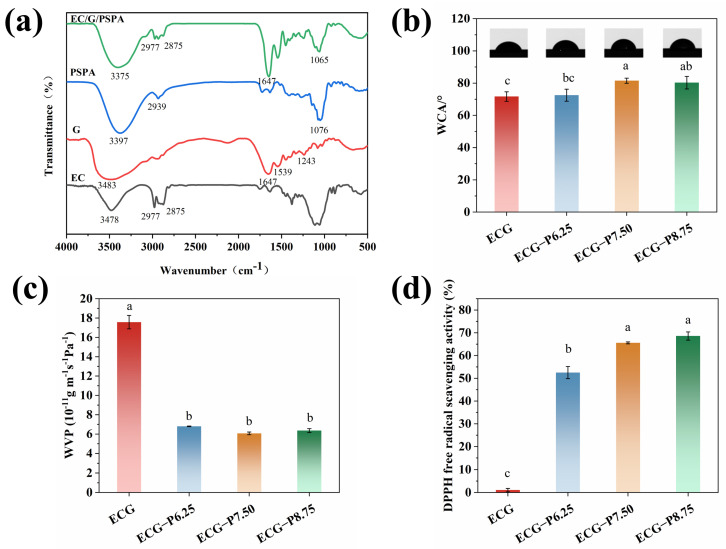
FTIR spectra (**a**), WCA (**b**), WVP (**c**), and antioxidant capacity (**d**) of the films. For graphs, the different letters (a–c) indicate that the samples are of significant difference (*p* < 0.05).

**Table 1 foods-13-00717-t001:** Color parameters of ECG−P7.50 nanofiber films and casting films when exposed to ammonia solution.

	Time (min)	Sample	L*	a*	b*	ΔE
Nanofiber films	0		84.44 ± 0.68 ^a^	2.99 ± 0.28 ^a^	5.37 ± 0.28 ^d^	-
1		85.29 ± 1.38 ^a^	−10.82 ± 1.00 ^b^	12.52 ± 1.28 ^c^	15.58 ± 1.54 ^c^
5		83.93 ± 1.66 ^a^	−14.56 ± 1.40 ^c^	23.79 ± 1.22 ^b^	25.45 ± 1.84 ^b^
10		85.77 ± 1.94 ^a^	−11.67 ± 0.06 ^b^	29.76 ± 1.19 ^a^	28.49 ± 1.00 ^a^
Casting films	0		59.17 ± 1.76 ^b^	12.71 ± 0.70 ^a^	1.02 ± 0.73 ^d^	-
1		62.17 ± 0.49 ^a^	7.52 ± 0.52 ^b^	7.50 ± 0.82 ^c^	8.84 ± 1.06 ^c^
5		62.38 ± 1.39 ^a^	1.98 ± 0.26 ^c^	16.00 ± 0.16 ^b^	18.72 ± 0.29 ^b^
10		62.75 ± 0.88 ^a^	−2.01 ± 0.51 ^d^	23.15 ± 1.33 ^a^	26.83 ± 0.67 ^a^

Different lowercase letters represent significant differences (*p* < 0.05).

**Table 2 foods-13-00717-t002:** TBARS value of pork sample and color parameters of ECG−P7.50 nanofiber film.

Time	TBARS	L*	a*	b*	∆E	Color
0	0.22 ± 0.012 ^a^	80.11 ± 1.50 ^bc^	7.63 ± 0.95 ^d^	6.65 ± 0.31 ^a^	—	
2	0.27 ± 0.003 ^b^	80.32 ± 0.53 ^bc^	3.45 ± 1.13 ^c^	6.85 ± 0.75 ^ab^	4.26 ± 1.12 ^e^	
4	0.34 ± 0.011 ^c^	82.47 ± 0.42 ^c^	0.95 ± 0.28 ^b^	8.07 ± 0.78 ^bc^	7.25 ± 0.45 ^d^	
6	0.48 ± 0.028 ^d^	79.20 ± 1.51 ^b^	−0.84 ± 1.10 ^a^	8.90 ± 0.37 ^c^	8.91 ± 1.03 ^cd^	
8	0.57 ± 0.036 ^e^	75.63 ± 2.01 ^a^	−0.31 ± 0.73 ^ab^	12.94± 0.99 ^d^	11.24 ± 0.13 ^b^	

Different lowercase letters represent significant differences (*p* < 0.05).

## Data Availability

The original contributions presented in the study are included in the article, further inquiries can be directed to the corresponding authors.

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
