# Peer review of "A Colorimetric Nanofiber Film Based on Ethyl Cellulose/Gelatin/Purple Sweet Potato Anthocyanins for Monitoring Pork Freshness"

_foods, 2024, doi:10.3390/foods13050717_

Round 1
Reviewer 1 Report
Comments and Suggestions for Authors
Dear Authors,
I have reviewed the paper, and although the topic is interesting, I believe that the paper needs improvement.
The introduction lacks flow and the whole paper it is challenging to follow.
The authors did not specify from which anatomical part of the pork the tested meat samples were taken. Since the study involves identifying antioxidant capacity, the fat content of the tested meat is very important. A pork piece with higher fat content will oxidize more quickly than one with lower fat content.
With regard to the research activity, perhaps the authors can explain the actual method of calculating encapsulation efficiency (line 149). The authors describe a colorimetric method, but the calculation relationship refers to the anthocyanin content on the film and it is used mass into equation. If they observed the sample at 530 nm why do they express the mass into EE equation?
The authors did not mention the SEM (Scanning Electron Microscopy) settings data (lines 156-160).
Line 224-225 the authors stated: "It can be seen that the pink to green color changes were easily captured by naked eyes for ECG-P6.25, ECG-P7.50 and ECG-P8.75 nanofiber films, while the light color of ECG P5.00 nanofiber film was difficult to be detected by consumers." Was there any sensory analysis involving human subjects? According to the manuscript and information referring to the informed consent statement, consumers were not involved in the research. In this case, how did the authors state that the light color was difficult for consumers to identify?
Taking into account the authors' statement "Oxidation is one of the major causes for meat spoilage, causing detrimental effects such as vitamin degradation and loss of essential fatty acids. Antioxidant film has the potential to help extend the shelf life of pork, as predicted." (lines 299-301), why did they not conduct microbiological tests to strengthen their claims and undertake a more comprehensive research approach?
Comments on the Quality of English Language
Moderate editing of English language required
Reviewer 2 Report
Comments and Suggestions for Authors
The article “A Colorimetric Nanofiber Film Based on Ethyl cellulose/ 2 Gelatin/Purple Sweet Potato Anthocyanins for Monitoring 3 Pork Freshness” by Peng Wen et al. is structured following the classic model for this type of material (Research Article), comprising four parts: Introduction, Materials and Methods, Results and Discussion, and Conclusions.
The submitted manuscript requires a thorough revision of English.
I list comments and suggestions for the authors.
Materials and methods
_Line 102 - In subchapter 2.2. must indicate under what conditions the PSPA should be stored if it is not used immediately.
_Line 111 - What are the buffer solutions and their concentrations?
_Line 118 - The sentence "The solution was sampled and then assessed at 517 nm by the spectrophotometer" is not understood. What do you understand by "solution was sampled"?
_Line 119 - Replace the sentence "The DPPH scavenging activity (%) was calculated by the following equation (1)" with "The DPPH free radical scavenging activity (%) was calculated by the following equation (1)"
_Equation (1) - The equation presented is not correct. It should be changed to "Scavenging activity (%) = ((A0-AA30)/A0)*100", where A0 is the absorbance of the control and A30 is the absorbance of the sample after 30 minutes.
I ask the authors to review the DPPH results in the results chapter.
_Line 139 - "The ammonia response of the nanofiber films was recorded by smart phone at 5 and 10 min" - what is the procedure to perform this task? How robust are these results? How do you guarantee the validity of these results?
_Line 2.6.1. Replace "Encapsulation Efficiency" with "Encapsulation Efficiency (EE)".
_Equation (3) - Remove the "%" symbol from the end of the equation.
_Equation (5) - What does the value "9.24" correspond to?
Results and discussion
_line 224 - Replace "It can be seen that the pink to green color" with "It can be seen in Fig. 2 that the pink to green color".
_Figure 4 - Add the legend to the graphs.
_Table 1 - Are the values corresponding to the "casting film" taken from the literature? If so, the references are missing. If not, explain where these values come from.
Comments on the Quality of English Language
It must be improved.
Round 2
Reviewer 1 Report
Comments and Suggestions for Authors
The paper has been improved.
Author Response
Thank you for your helpful comments and appreciation of our work.
Reviewer 2 Report
Comments and Suggestions for Authors
I am grateful to the authors for their dedication in reviewing the manuscript. The final version is excellent and deserves to be widely shared.
Author Response
Thank you for your professional review work, constructive comments, and valuable suggestions on the manuscript.